# Advances in Pediatric Surgery Simulation-Based Training

**DOI:** 10.3390/children11010034

**Published:** 2023-12-28

**Authors:** Laquanda T. Knowlin, Nicholas M. B. Laskay, Nehemie P. Jules, Jakub Godzik, Todd P. Chang, Ryan G. Spurrier

**Affiliations:** 1Division of Pediatric Surgery, Children’s Hospital Los Angeles, 4650 Sunset Blvd., Mailstop #175, Los Angeles, CA 90027, USA; 2Las Madrinas Simulation Research Laboratory, Children’s Hospital Los Angeles, Keck School of Medicine, University of Southern California, Los Angeles, CA 90027, USA; 3Department of Neurosurgery, University of Alabama at Birmingham, Birmingham, AL 35294, USA; 4Division of Emergency and Transport Medicine, Children’s Hospital Los Angeles, Los Angeles, CA 90027, USA

**Keywords:** pediatric surgery, simulation training, education, neonatal

## Abstract

Pediatric surgery is the diagnostic, operative, and postoperative surgical care of children with congenital and acquired anomalies and diseases. The early history of the specialty followed the classic “see one, do one, teach one” philosophy of training but has since evolved to modern methods including simulation-based training (SBT). Current trainees in pediatric surgery face numerous challenges, such as the decreasing incidence of congenital disease and reduced work hours. SBT consists of several modalities that together assist in the acquisition of technical skills and improve performance in the operating room. SBT has evolved to incorporate simulator models and video gaming technology, in parallel with the development of simulation in other surgical and non-surgical pediatric fields. SBT has advanced to a level of sophistication that means that it can improve the skills of not only pediatric surgery trainees but also practicing attending surgeons. In this review, we will discuss the history of pediatric surgery, simulation in pediatric surgery training, and the potential direction of pediatric surgical simulation training in the future.

## 1. Introduction

Pediatric surgery involves the diagnostic, operative, and postoperative surgical care of children with congenital and acquired anomalies and diseases [1]. There is no universal definition of a pediatric surgeon, as some are recognized through specific board certification while others have developed a niche based on clinical experience alone [2]. The journey to become a surgeon that operates on this special population is tedious but rich in history. In this review, we will outline the history of pediatric surgery as a subspeciality through the lens of earlier training including advancements in diversity, summarize current training methodologies using simulation, and discuss the next steps in integrating simulation best practices into pediatric surgery training in the future. 

### 1.1. The History of Pediatric Surgery Training

The history of surgery dates back many centuries and stretches across many continents [3]. In ancient times (1500–3500 BC), doctors and surgeons were predominantly female in European and Middle Eastern countries [3]. Most women learned through apprenticeships from their fathers or husbands before taking over the practice at the time of death or disguised themselves as men to obtain medical training. Even later during the Middle Ages, women were discouraged from practicing medicine [4]. In the late 20th century, earlier surgeons who desired to care for children either traveled to operate under the guidance of pioneers at the Children’s Hospital (now Boston Children’s Hospital, USA) or the Hospital for Sick Children (now Great Ormond Street, UK) before leaving and establishing pediatric surgery departments elsewhere [5,6,7]. These flagship hospitals evolved to become future sites for pediatric surgeon training under an apprenticeship model. Some of the first operations were on fully anesthetized children and included correction of intussusception by William E. Ladd and imperforate anus by John C. Warren [8,9]. In the literature, other techniques for learning such as repetitive “simulated” practice with vivisection in dogs have been described by surgeon Mark Ravitch, who devised the mucosal proctectomy, colectomy, and ileoanal anastomosis. Repetitive “simulated” practice on animals was the method by which a laboratory assistant named Vivien Thomas enabled the creation of the Blalock–Thomas–Taussig shunt for tetralogy of Fallot before it was performed in actual humans [10,11]. The use of animals to simulate pediatric surgical care was the method by which surgical practice could be carried out across all people. 

As more pediatric surgeons were being trained, the field itself yearned for recognition as a separate entity under general surgery [5,12]. The Royal College of Surgery officially recognized pediatric surgery as a separate subspecialty in 1963 [5]. Surgeons practice as pediatric surgeon consultants after two years of basic surgery training followed by several years of subspecialty clinical experience and successful completion of a written examination [5]. It took a few decades before the specialty received recognition from the American Board of Surgery in 1973, with Canadian surgeon Harvey Beardmore leading the campaign [12]. North American pediatric surgery fellowships developed to become 2 years in length following the completion of 5 to 7 years of general surgery training [13]. Board certification became available after the passage of written and oral examinations in both general and pediatric surgery. However, certification using the surgical performance evaluation of simulated performance was not part of the board certification process. Most countries followed these models with recognition of pediatric surgeon status after years of training and final examinations [2]. In addition, some countries require surgical performance evaluation or non-surgical pediatric experience [14,15,16].

### 1.2. Current Challenges in Training Pediatric Surgeons

Many training programs for pediatric surgery have been developed worldwide, but there are several challenges in the training of pediatric surgeons [17]. These include limited exposure to rare and complex pediatric cases even in large teaching programs, the declining global incidence of congenital disease requiring surgical intervention, regulations on duty hours by governing training boards, and limitations on the types of surgical cases as a result of healthcare system resources, such as the establishment of specific institutes that collect only certain cases at a training hospital [14]. Gone are the days of the “see one, do one, teach one” philosophy in the operating room [18]. As an adjunct, the utilization of simulation-based training (SBT) has evolved over the past decade for both general surgeons and pediatric surgeons to provide supplementary experiences. 

## 2. Methods

A broad search of the PubMed database was performed to identify articles that described pediatric surgery history, training, and simulation-based education. Search terms included “pediatric or paediatric” and “surgery” and “training” and “education” and “simulation”. Titles and abstracts were screened for content relative to the narrative review.

### 2.1. The Onset of Simulation-Based Training in Pediatric Surgery 

Simulation-based training in pediatric surgery is still nascent. While simulation has been embraced by other surgical subspecialties such as urology, orthopedics, and ENT [19,20,21], pediatric surgery lags behind general surgery in simulation. As with many areas of medicine, the field focused initially on the adult population. For example, there are many more trainers available today simulating adult anatomy than pediatric or infant versions. It is common practice in simulation to want to emulate as much realism, or fidelity, as possible. High physical and anatomical fidelity is particularly important in surgical simulators, even more than non-surgical simulators that emphasize functional fidelity. The level of fidelity can range from low to high, but the optimal selection of fidelity is based on the learning objectives. Surgery in pediatric patients requires operating in very small, narrow spaces of the thoracic, abdominal, and pelvis (genitourinary) compartments compared to adults. For example, the peritoneal cavity in an infant is far smaller than in an adult, and surgical instruments, hand movements, and strategies have been formulated to accommodate for the restricted space. As with adult general surgery, interest in minimally invasive surgery has become more common and is commonly taught as laparoscopic surgery, instead of laparotomy surgery [22]. Pediatric space constraints for surgeons can make laparoscopic techniques far harder [1]. The individual development of these skills can be accomplished through low-fidelity laparoscopic trainers for basic skills that trainees can take home [23,24]. Once these skills are mastered, other modalities of simulation can be utilized to prepare trainees for more complex operative procedures. In general, SBT in surgery can include cadaver and vivisection animal models, inanimate task trainers, and innovative strategies like digital or 3D-printed models. 

### 2.2. Cadaver and Vivisection

The first surgical “simulations” were operations on live animals or cadaveric (animal and human) tissue [25]. Vivisection, a controversial topic, is the performance of complex procedures on a live anesthetized animal that closely emulates an actual operating setting [26]. Over time, ethical standards have shifted away from these modalities, in animals less than humans, but there are still known advantages to training on live subjects [27,28]. Live tissues have the obvious benefits of tactile and visual fidelity during handling [29]. Trainees are able to practice basic surgery skills and more complex procedures such as laparoscopic duodenal atresia and diaphragmatic hernia repair [28,30,31]. On the other hand, cadaveric tissue and the vivisection of animals can be costly and logistically difficult to store and curate [27,32]. Before the COVID-19 pandemic, U.S. fellows were able to attend an annual minimally invasive surgery course at Northwestern University that utilized inanimate 3D-printed models and cadaver tissue [33]. The availability of vivisection varies based on the institution’s association with nearby veterinarian schools. Internationally, vivisection with porcine and smaller animal labs for laparoscopic skill development has been performed in pediatric surgery programs [30,34]. In addition to emulating a small operative internal cavity space, anesthetized animals can be monitored on the operating room table as part of a high-fidelity simulation experience [34]. Unfortunately, the animals either die during the procedure or are euthanized afterward. If handling animal tissue is not an option, other options exist in pediatric surgery education.

### 2.3. Simulation Models and Trainers

For more realism, pediatric surgery has turned to higher fidelity simulators to improve technical skills for bedside and operative index cases in pediatric surgery [35]. Higher physical fidelity more closely approximates the feel, texture, heft, and bounce of an intestine or other abdominal organ. These simulators can be created with three-dimensional (3D) printing or inexpensive items [35,36,37]. While these are very innovative solutions for surgical training and assessment, surgical simulators have two current issues: availability and validity evidence. Joosten et al. wrote a systematic review examining the validity of simulators mentioned in the literature [38]. Of the 38 models reported in the literature, only 5 models are commercially available and 11 models were replicable based on the article description. The studies for these models evaluated face and content validity, the subjective view of how realistic a simulation is, in comparison to construct validity, and objective sense to the extent to which a simulation provides an accurate representation of a real task [39]. Establishing validated simulation models is critical [40] and often carried out with assessments that measure discriminant (construct) validity between students, residents, and attending consultant surgeons. For example, a simulation assessment that can distinguish low student performance vs. high attending consultant performance is likely to have appropriate validity evidence. However, validity evidence beyond discriminant validity for pediatric simulators is rare. Only one study to date discusses predictive validity assessments in which performance in simulation correlates well with later performance in the operating room [41,42]. Despite the number of models described in the literature, more validity and interventional studies are required to provide recommendations for best practices [43]. 

### 2.4. Virtual, Augmented, and Mixed Reality

Another simulation innovation modality seen recently in pediatric surgery is the use of virtual (VR), augmented (AR), and mixed reality (MR). Video gaming technology provides a virtual environment for participants to practice clinical and non-technical skills without harming actual patients. Limitations to this modality are cost, cybersickness, and the degree of fidelity [44]. Technology has become more advanced as VR and AR can now be combined with three-dimensional (3D) printing to aid in preoperative planning for complex surgeries [45]. This requires uploading an individual’s radiographic image (i.e., CT, MRI, or 3D ultrasound) into the virtual environment for the surgeon to manipulate. Several studies have examined the use of 3D VR or AR for preoperative planning in spinal, orthopedic, OMFS, and oncological surgeries [46,47,48,49]. Outcomes include a reduction in operative times and improved communication amongst OR team members. Pelizzo et al. wrote about developing the first use of a 3D VR head-mounted display in preparation for surgical removal of congenital lung malformations in children [50]. For the study, a highly detailed 3D model of the malformation was created from a segmented CT image. This imaging is uploaded to a head-mounted device worn by the surgeon who is able to alter the image and share a point-of-view correspondence with the rest of the team for discussion [46]. Hopefully, in the future, this type of training can expand to complex additional pediatric-specific conditions. 

## 3. The Future Direction of the Simulation-Based Training of Pediatric Surgeons

As pediatric surgical simulation training continues to evolve, the future needs of the field should include the creation of more models tailored to pediatric surgery, the expansion of video gaming technology to clinical areas of limited training exposure, and more implementation of simulation-based curricula.

Previous reviews have noted that very few pediatric surgical simulators are readily available for purchase, but some literature examples enable departments and hospitals to develop their own [35]. Now that 3D printers have become more commercialized, some hospitals have 3D printing programs or institutes for clinical and research purposes. It is possible to print anatomic structures—like bones to internal organs—that can be handled during surgical procedures. Engineers and trained technicians can therefore develop products to meet the haptic and tactile needs of surgeons (Figure 1). A hybrid of 3D printing and cadaver tissue has even been used [30].

### 3.1. Technology Expansion

The expansion of video gaming technology also enables training in areas of limited exposure for pediatric surgery trainees. These are low-frequency, high-stakes events. One such area is pediatric trauma education. Pediatric trauma is the leading cause of mortality in children [51]. It is essential to prepare providers, surgeons, and non-surgeons to take care of this special population to improve outcomes. However, specific simulators for trauma are very limited, especially for pediatric and infant patients. Moulage, special makeup, is often used to simulate critical injuries on non-trauma-designated manikins (Figure 2a,b). Trauma simulators can range from USD 14,000 to USD 200,000 depending on the size of the manikin and the level of desired fidelity [52,53,54]. A lower-cost simulation solution is digital. Several companies are creating simulation training through VR to train providers with minimal exposure to pediatric trauma [55]. Specialized topics include mass casualty situations in both the military and civilian populations. It provides a psychologically safe environment to practice making life-saving decisions in preparation for actual events. Taking it a step further, augmented reality (AR) decision-making systems with algorithms for trauma activations have been designed in New Zealand, with expanded use in Asia and the Middle East [56]. Collaboration among adult and pediatric surgeons can expand this further and help reduce morbidity and mortality outcomes in children around the world. The surgical field is shifting to implementing artificial intelligence (AI) which encompasses machine learning, to offer feedback on trainee surgical performance after video review [57]. As surgical fields such as neurosurgery, urology, and some general surgery subspecialties look to artificial intelligence, the question remains if pediatric surgery will follow suit [58,59,60]. Similar to 3D VR, AI has been proposed for pre-operative planning with a focus on visualization [61]. However, there is still work that needs to be done as far as reducing implementation and production costs, providing clear evidence that this technology is superior to the current standard, and addressing intrinsic technology limitations (i.e., cybersickness, head-mounted devices, and power for graphic processors) [56]. Future technologies are likely to improve haptic sensations in VR and digital simulations, building on hybrid models that combine tactile sensations with digital solutions to achieve learning objectives. Haptics are not optimized for current digital simulations, despite the importance of haptic feedback in pediatric surgical procedures. We predict that haptic technology will enhance future training, which will fully enable digital and distance-based simulations for pediatric surgeons [62]. In addition, as the future becomes even more technology-driven, we must not forget the importance of pediatric surgery training in countries with limited resources or technological access (Figure 3). 

### 3.2. The Widespread Implementation of Simulation-Based Curricula

The idea of simulation-based training was to build on technical skills that will be transferred to improve performance in the operating room and/or to fill in the education gaps for limited exposure to certain pediatric surgical cases. In addition to evaluating technical skills, simulation has been shown to be superior to didactic training courses for non-technical skills such as leadership, communication, situational awareness, and decision-making [63]. This is important as up to 35% of total adverse events in children are reported during the perioperative period, and communication is thought to be a factor in 43% of errors made in surgery [64,65]. A survey of pediatric surgical trainees as part of a simulation program in France reported inadequate training in the area of non-technical skills [66]. More pediatric surgery programs have used simulation as part of boot camps or developed simulation-based curricula as part of trainee development [32,67,68]. With the implementation of minimally invasive surgery three decades ago, MIS procedures have become more integrated into the pediatric surgery training curriculum. From 2004 to 2016, there was a 30% increase in the average number of MIS cases per fellow in Canada and the United States with variation in exposure among trainees [69]. In contrast, 67% of pediatric surgery trainees from European countries responded to a survey of the challenges in their program performing of MIS procedures, and fewer than 5 out of 25 pediatric MIS procedures were performed by at least 50% of trainees [22]. Like in Northern America, there was great variability in training and exposure in Europe as years spent in a general surgery department led to a greater number of higher complex procedures performed compared to years in pediatric surgery training [22]. Some fellowship programs have developed and implemented MIS programs to help mitigate this weakness in training fellows. Recently, Bailez et al. wrote about the development of a minimally invasive surgery training program, onsite and telesimulated, with low-cost models in Argentina that has led to surgical efficiency and an increase in the complexity of cases performed [70]. In low- to middle-income countries (LMICs), there is a greater need to train more pediatric surgeons, and the method of training varies per country [71,72]. Almost half of the population in LMICs are children under the age of 15 [73]. Compared to high-income countries (HICs), LMICs do not have enough resources at each site, but substantial improvements have been made [74]. Education methods include collaborations between countries with different resources. High-income countries have sent visiting teams to teach workshops on complex procedures, acting as mentors to trainees and, through advocacy after returning home, helping bring awareness to the training needs of LMIC [75]. Other training methods used in the past include international pediatric surgery fellowships and fellow exchange programs in HICs to further improve the availability of diverse pediatric surgery talent [72,76,77]. A fellow exchange program between Montreal Children’s Hospital (Canada) and Bethany Kids of Kijabe Hospital (Kenya) resulted in more frequent exposure to neonatal, MIS, and vascular procedures for Kenyan trainees [77]. Reported difficulties from Kenyan surgical trainees were obtaining a training license and the financial burden of the cost of living in HICs. As of today, some pediatric surgery training centers in Africa still have limited exposure to trauma, burn, and minimally invasive surgery [78]. A recent survey of surgery residents in 11 Sub-Saharan African countries apart of the College of Surgeons of East, Central, and South Africa (COSECSA) strongly believed that simulation was valuable and should be part of their training program [79].

### 3.3. The Metric-Based Evaluation of Surgical Trainees in Simulation Training

Prior to the introduction of anesthesia in the 19th century, the early metric in the assessment of skills for surgeons was operative time length. This type of evaluation gave no indication of the quality of performance or feedback. Metric-based assessment in surgery has evolved with the development of the Objective Structured Assessment of Technical Skills (OSATS) which is similar to the Objective Structured Clinical Examinations (OSCE) commonly used to evaluate healthcare students [80]. Since its first creation, the OSATS has been modified in different surgical fields such as CT surgery and gynecology [81,82]. Our institution has modified a version of the OSATS for pediatric surgery simulations that will be included in semi- and annual performance reviews. Although the OSATS is widely accepted in surgery, the greatest issues are a lack of objectivity, it does not meet the extrapolation criteria for Kane’s validity framework, and the inability to differentiate competencies for more experienced trainees due to the ceiling effect of the rating scale [83,84]. The development of a successful metric might appear to be simple, but more attention needs to be given to breaking down procedure-specific tasks into their essential components, strictly defining what differentiates optimal from suboptimal performance, and including non-technical skills. The combination of an assessment tool with video recordings strengthens the evaluation of trainee performance and enables evaluation for discriminative validity [85].

## 4. Conclusions

The field of pediatric surgery training has evolved to produce specialized surgeons capable of performing complex operations on the smallest of children. From repetitive training involving both direct observation in the operating room to state-of-the-art simulation-based training, today’s pediatric surgical trainees have a much broader exposure to pediatric diseases and procedures than their predecessors. The ongoing innovations in pediatric surgery simulation and the worldwide implementation of simulation curricula are necessary to train the increasingly diverse future generations of pediatric surgeons.

## Figures and Tables

**Figure 1 children-11-00034-f001:**
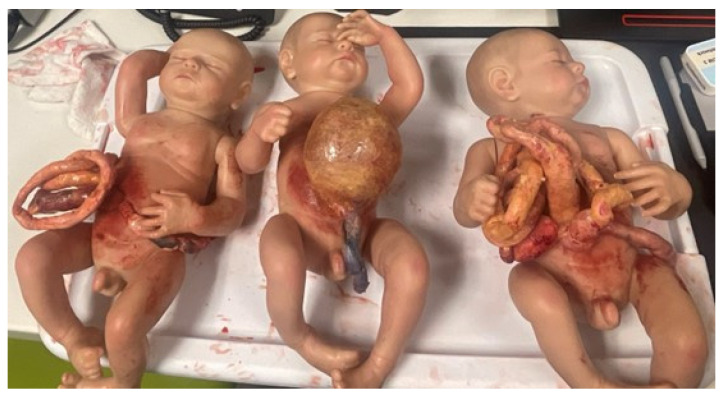
Neonatal Abdominal wall defect models created for the Pediatric Surgery Fellow Workshop at Las Madrinas Simulation Center (Los Angeles, CA, USA).

**Figure 2 children-11-00034-f002:**
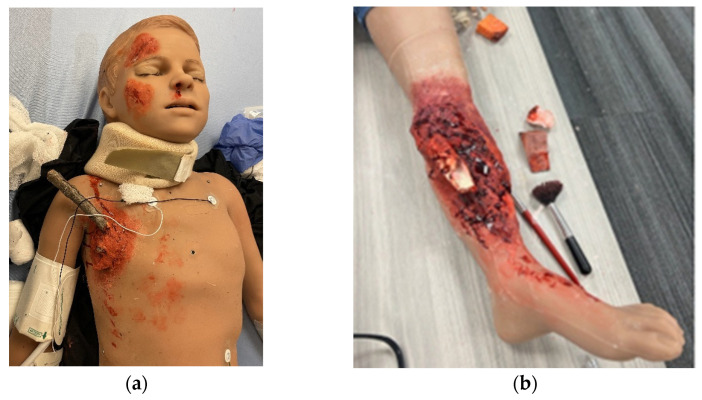
Trauma moulage to simulate critical injuries on a pediatric medical simulator (Pediatric HAL, Gaumard Scientific) at Las Madrinas Simulation Center (Los Angeles): (**a**) penetrating chest trauma and (**b**) open orthopedic fracture.

**Figure 3 children-11-00034-f003:**
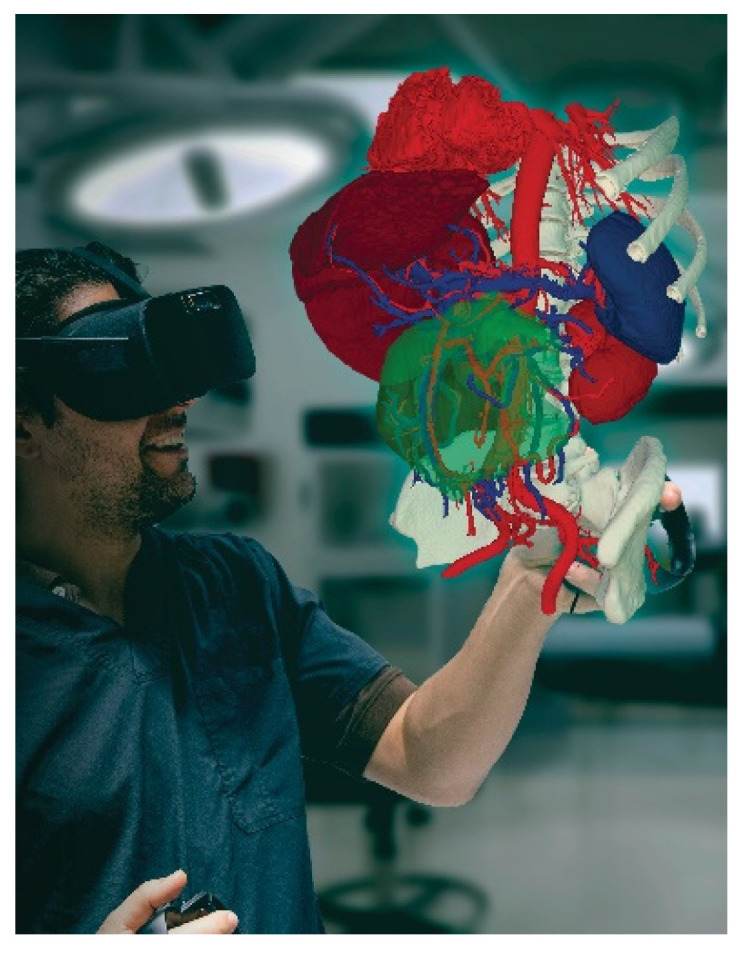
Preoperative planning using 3D virtuality reality software ImmersiveView 5.0 by ImmersiveTouch (Chicago, IL, USA).

## Data Availability

Not applicable.

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
