# Peer review of "Advances in Pediatric Surgery Simulation-Based Training"

_children, 2023, doi:10.3390/children11010034_

Round 1

Reviewer 1 Report

Comments and Suggestions for Authors

This paper provides a comprehensive and well-organized review of the history and current state of pediatric surgical training, with a specific focus on simulation-based methodologies. The integration of historical context, contemporary challenges, and future prospects demonstrates a thorough understanding of the subject matter. The inclusion of specific studies and examples enhances the paper's academic rigor. While the paper is generally well-written, there are instances where more concise language could improve readability. Additionally, the paper could benefit from more explicit transitions between sections to enhance overall coherence.

Overall, the paper makes a valuable contribution to the literature on pediatric surgical training and simulation.

Summary of the Manuscript:
The manuscript explores the history and current state of pediatric surgery training, focusing on the evolution of the field, challenges in training, and the role of simulation-based training (SBT). It covers the historical context of pediatric surgery, the emergence of dedicated pediatric surgeons, and the evolution of training methodologies. The paper discusses the challenges in pediatric surgery training, emphasizing the need for exposure to rare cases and the limitations of the traditional "see one, do one, teach one" approach. The central theme revolves around the incorporation of SBT in pediatric surgery training, highlighting the nascent stage of its implementation and the challenges associated with fidelity, availability, and validation of simulators. The manuscript concludes by discussing the future of pediatric surgery training, including the creation of tailored models, the expansion of video-gaming technology, and the widespread implementation of simulation-based curricula.

Strengths:
Comprehensive Historical Overview: The manuscript provides a thorough historical overview of pediatric surgery, tracing its roots from ancient times to modern-day, highlighting key figures, and pivotal moments.
Identification of Challenges: The manuscript effectively identifies the challenges in pediatric surgery training, including limited exposure to rare cases, declining global incidence of congenital diseases, and regulatory constraints.
In-depth Exploration of Simulation: The manuscript delves into the world of simulation-based training, discussing various modalities such as cadaver and vivisection, simulation models and trainers, and virtual, augmented, and mixed reality. It appropriately recognizes the importance of fidelity in pediatric surgery simulation.

Weaknesses:
Organization: The manuscript could benefit from improved organization. Consider restructuring sections for better flow, such as separating historical aspects from current challenges and simulation methods.
Balance in Historical Section: While the historical context is important, ensure that the balance between historical information and the focus on current and future aspects of pediatric surgery training is maintained.

Recommendations:
Structural Enhancements: Reorganize the content for a smoother flow, ensuring a logical progression from historical aspects to current challenges and future directions.
Expand on Simulation Validation: Elaborate on the importance of validating simulators in pediatric surgery. Provide more information on the criteria for validation and its implications for training.
Foster International Collaboration: Emphasize the importance of international collaboration in pediatric surgery training, particularly in low to middle-income countries. Provide more concrete examples and strategies for collaboration.

Comments on the Quality of English Language

see the above.

Author Response

Thank you for giving us the opportunity to submit a revised draft of the manuscript “Advances in Pediatric Surgery Based Training” for publication in special issues Advances in Pediatric and Neonatal Simulation for Children. We appreciate the time and effort that you dedicated to providing feedback on our manuscript and are grateful for the insightful comments on and valuable improvements to our paper. We have incorporated most of the suggestions. Those changes are highlighted within the manuscript. Please see below, in blue, for a point-by-point response to the reviewers’ comments and concerns. All page numbers refer to the revised manuscript file with tracked changes.

1. Structural enhancements: Reorganize the content for a smoother flow, ensuring a logical progression from historical aspects to current challenges and future directions.

As suggested by the reviewer, we have reorganized the manuscript.

2. Expand on simulation validation: Elaborate on the importance of validation simulators in pediatric surgery. Provide More information on the criteria for validation and is implications for training.

As suggested by the reviewer we have added the following to section 2.2 Simulation models and trainers. (Page7)

“The studies for these models evaluated face and content validity, the subjective view of how realistic a simulation is, in comparison to construct validity, objective sense to the extent to which a simulation provides an accurate representation of a real task.”

“For example, a simulation assessment that can distinguish low student performance vs high attending consultant performance is likely to have appropriate validity evidence.”

3. Foster International Collaboration: Emphasize the importance of the international collaboration in pediatric surgery training, particularly in low to middle-income countries. Provide more concrete examples and strategies for collaboration.

As suggested by the reviewer, we have added the following to section 3.2 Widespread implementation of simulation based curricula. (Pages 12 and 13)

“High income countries have sent visiting teams for teaching workshops on complex procedures, act as mentors to trainees, and, through advocacy after returning home, help bring awareness to the training needs of LMIC. Other training methods used in the past include international pediatric surgery fellowships and fellow exchange programs in HICs and to further improve the availability of diverse pediatric surgery talent. A fellow exchange program between Montreal Children’s Hospital (Canada) and Bethany Kids if Kijabe Hospital (Kenya) resulted in more frequent exposure to neonatal, MIS, and vascular procedure for the Kenyan trainees. Reported difficulties from Kenyan surgical trainees were obtaining a training license and the financial burden for cost of living in a HICs. “

Reviewer 2 Report

Comments and Suggestions for Authors

This manuscript is a review article that includes the history of pediatric surgery, simulation in pediatric surgery training, and the potential direction of pediatric surgical simulation training in the future.  The authors contend that simulation-based training has advanced to a level of sophistication that it can improve the skills of not only pediatric surgery trainees but also practicing attending surgeons.  This is a well-written and comprehensive review of the existing literature on simulation-based training for the field of pediatric surgery.  A weakness of the study is the lack of information related to how best to evaluate trainees that are enrolled in simulation-based curricula.  I have some minor suggestions:
•    Despite a relatively comprehensive review of the history of pediatric surgery, there is an important omission related to Dr. William Ladd’s contribution to the children affected by the Halifax Explosion.  Many feel that this important historical event led to the birth of modern pediatric surgery.
Nance ML. The Halifax disaster of 1917 and the birth of North American pediatric surgery. J Pediatr Surg. 2001;36(3):405-408.

•    Section 2.1 – Define “vivisection”.

•    Section 2.1 – What is the status of the annual minimally invasive surgery course at Northwestern University?

•    A very well written review article.  Excellent organization and information.  I would consider adding a section 3.3 related to "evaluation of trainees using simulation-based training methods". There is a rather abrupt transition between section 3.2 to the conclusions.  Add in a section 3.3 that outlines the evaluation of trainees that are enrolled in simulation-based curricula.

Author Response

Thank you for giving us the opportunity to submit a revised draft of the manuscript “Advances in Pediatric Surgery Based Training” for publication in special issues Advances in Pediatric and Neonatal Simulation for Children. We appreciate the time and effort that you dedicated to providing feedback on our manuscript and are grateful for the insightful comments on and valuable improvements to our paper. We have incorporated most of the suggestions. Those changes are highlighted within the manuscript. Please see below, in blue, for a point-by-point response to the reviewers’ comments and concerns. All page numbers refer to the revised manuscript file with tracked changes.

1. Despite a relatively comprehensive review of the history of pediatric surgery, there is an important omission related to Dr. William Ladd’s contribution to the children effect by the Halifax Explosion. Many feel that this important historical event led to the birth of modern pediatric surgery.

Thank you for this history fact. However, other reviewers would like to remove some of the history review of pediatric surgery from the manuscript.

2. Section 2.1- Define “vivisection”

As suggested by the reviewer, we have added a definition of vivisection to section 2.1. Cadaver and Vivisection. (Page 6)

“ Vivisection, a controversial topic, is the performance of complex procedures on a live anesthetized animal that closely emulates an actual operating setting.“

3. Section 2.1- What is the status of the annual minimally invasive surgery course at Northwestern University?

The minimally invasive surgery course will be held at the International Pediatric Endosurgery Group in June 2024. It is being organized by Northwestern University/Lurie Children’s and an announcement will be coming out soon.

4. A very well written review article. Excellent organization and information. I would consider a section3.3 related to “evaluation of trainees using simulation-based training methods.” There is a rather abrupt transition between section 3.2 to the conclusion. Add in a section 3.3 that outlines the evaluation of trainees that are enrolled in simulation-based curricula.

As suggested by the reviewer, we have added a section 3.3 on Metric-based evaluations of surgical trainees in simulation training. (Pages 13 and 14)

“Prior to the introduction of anesthesia in the 19th century, the early metric in assessment of skills for surgeons was operative time length. This type of evaluation gave no indication of the quality of performance. One of the goals of surgical simulation-based training is that will translate to improvement in performance or feedback. Metric-based assessment in surgery had evolved with the development of the Objective Structured Assessment of Technical Skills (OSATS) which is similar to the Objective Structured Clinical Examinations (OSCE) commonly used to evaluate healthcare students. Since its first creation, the OSATS has been modified in different surgical fields such as CT surgery and gynecology. Our institution has modified a version of the OSATS for pediatric surgery simulations that will be included in semi- and annual performance reviews. Although the OSATS is widely accepted in surgery, greatest issues are lack of objectivity, it does not meet the extrapolation criteria for Kane’s validity framework, and the inability differentiate competencies for more experience trainees due to ceiling effect of rating scale. Development of a successful metric might appear to be simple but more attention needs to be given to breaking down procedure specific task into its essential components, strictly defining what differentiates optimal from suboptimal performance, and include non-technical skills. The combination of an assessment tool with video recordings strengthens the evaluation of trainee performance and enable evaluation for discriminative validity.”

Reviewer 3 Report

Comments and Suggestions for Authors

1. In principle, paediatric surgery is not just a subspecialty of general surgery. Such a message is totally unacceptable to the reader. The basic principles of paediatric surgery training are essentially the same as those of paediatric surgery, but the treatment of certain diseases and congenital disorders differs significantly. In addition, surgical treatment relies on the physiological characteristics and responses of children, which is not the same as in the adult population.

2. Clinical experience in surgical treatment is mostly in the limited field of surgery or pathology. Comment, demonstrate, define.

3. The chapter on the history of training is unnecessary.

4. The chapters on how to demonstrate each type of training are too long, and to some extent irrelevant. The authors should be more specific about the importance and the way training will be done in the future, virtual reality and training and the pitfalls it brings.

5. What does communication skills training mean from the point of view of paediatric surgery, or how is this part of the paediatric surgeon's education organised?

Comments on the Quality of English Language

-

Author Response

Thank you for giving us the opportunity to submit a revised draft of the manuscript “Advances in Pediatric Surgery Based Training” for publication in special issues Advances in Pediatric and Neonatal Simulation for Children. We appreciate the time and effort that you dedicated to providing feedback on our manuscript and are grateful for the insightful comments on and valuable improvements to our paper. We have incorporated most of the suggestions. Those changes are highlighted within the manuscript. Please see below, in blue, for a point-by-point response to the reviewers’ comments and concerns. All page numbers refer to the revised manuscript file with tracked changes.

1. In principle, paediatric surgery is not subspeciality of general surgery. Such a message is totally unacceptable to the reader. The basic principles of paediatric surgery training are essentially the same those of paediatric surgery, but the treatment of certain diseases and congenital disorders differs significantly. In addition, surgical treatment relies on the physiological characteristics and responses of children, which is not the same as adult population.

Pediatric surgery is a subspeciality of general surgery in the United States. We have made changes to accommodate a more global definition. (Pages 2 and 3)

“Pediatric surgery is the diagnostic, operative, and postoperative surgical care for children with congenital and acquired anomalies and diseases.”

2. Clinical experience in surgical treatment is mostly in the limited field of surgery or pathology. Comment, demonstrate, define.

Thank you for your comment. This has been removed from the manuscript.

3. The chapter on the history of training is unnecessary.

Thank you for your comment. We feel the history of pediatric surgery training is necessary to show readers how surgical training has changed over time and includes simulation.

4. The chapters on how to demonstrate each type of training are too long and to some extent irrelevant. The authors should be more specific about the importance and the way training will be done in the future, virtual reality and training and the pitfalls it brings.

Thank you for your comment. Section 2 discusses the current types of simulation modalities that have been used for pediatric surgery training and Section 3 is specific to the future direction of simulation-based training. We have added the following  to section 3.1 Technology Expansion (Page 10)

“Future technologies are likely to improve haptic sensations in VR and digital simulations, building on hybrid models that combine tactile sensations with digital solutions to achieve learning objectives. Haptics are not optimized for current digital simulations, despite the importance of haptic feedback in surgical procedures. We predict that haptics technology will enhance future training that fully enable digital and distance-based simulations for pediatric surgeons.”

5. What does communication skills training mean from the point of view of paediatric surgery, or how is this part of the paediatric surgeon’s education organized?

As suggested by the reviewer, we have added to section 3.2 Widespread implementation of simulation-based curricula the following about non-technical skills such as communication to the manuscript. (Pages 11 and 12)

“In addition to evaluating technical skills, simulation has been shown to be superior to didactic training courses for non-technical skills such as leadership, communication, situational awareness, and decision-making. This is important as up to 35% of total adverse events in children are reported during the perioperative period and communication is thought to be a factor in 43% of errors made in surgery. A survey of pediatric surgical trainees as part of a simulation program in France reported inadequate training in the area of non-technical skills.”

Reviewer 4 Report

Comments and Suggestions for Authors

In this review, the authors discuss the history of pediatric surgery, simulation in pediatric surgical training, and the possible direction of pediatric surgical simulation training in the future.

I read the report with great interest. My concerns are as follows:

1. I disagree with the first sentence in the summary and introduction - Pediatric surgery is a subspecialty of general surgery that specializes in the diagnosis and operative management of surgical diseases affecting children. In my opinion, a more propriate deffinition would be - Pediatric surgery is defined as the diagnostic, operative, and postoperative surgical care for children with congenital and acquired anomalies and diseases, be they developmental, inflammatory, neoplastic or traumatic. The scope of this discipline would focus especially on surgical problems in utero, infancy, childhood, adolescence, and sometimes, young adulthood.

2. The main objectives of this review are still unclear as to what the search strategy for the literature was, what keywords were used and what databases were examined. This is important information that any review should include.

3. The paragraph ''1.1 History of pediatric surgery and training'' is beyond the scope of this review. The focus of this review should be on advances in simulation-based training in pediatric surgery. About 30% of the text of the article is a very detailed history which should be given in a short paragraph.

4. As we live in the era of minimally invasive surgery, much more attention and space should be given to minimally invasive training and the possible modalities. Minimally invasive simulators should be presented in much more detail.

5. The authors should examine the current situation in training the skills required for the different areas of pediatric surgery and compare them between Europe, Asia and America... For example, the authors should comment on the current situation of training in minimally invasive surgery and compare the experiences of trainees in paediatric surgery in Europe, America, Asia, etc.

6. In addition, the development of instruments specifically for the paediatric population should be described, which the authors did not address.

7. Finally, this review is very short (usually at least 4000 words are required, excluding references and explanations). Moreover, since many similar systematic reviews have been published on this topic, I do not see the scientific value of this simple narrative review.

Comments on the Quality of English Language

Minor editing of English language required

Author Response

Thank you for giving us the opportunity to submit a revised draft of the manuscript “Advances in Pediatric Surgery Based Training” for publication in special issues Advances in Pediatric and Neonatal Simulation for Children. We appreciate the time and effort that you dedicated to providing feedback on our manuscript and are grateful for the insightful comments on and valuable improvements to our paper. We have incorporated most of the suggestions. Those changes are highlighted within the manuscript. Please see below, in blue, for a point-by-point response to the reviewers’ comments and concerns. All page numbers refer to the revised manuscript file with tracked changes.

1. I disagree with the first sentence in the summary and introduction- Pediatric surgery is a subspecialty of general surgery that specializes in the diagnosis of operative management of surgical disease affecting children. In my opinion, a more propriate definition would be- Pediatric surgery is defined as the diagnostic, operative, and postoperative surgical care for children with congenital and acquired anomalies and diseases, be they developmental, inflammatory, neoplastic or traumatic. The scope of this discipline would focus especially on surgical problems in utero, infancy, childhood, adolescence, and sometimes, young adulthood.

As suggested by the reviewer, we have added a more appropriate definition of pediatric surgery to the abstract and introduction (pages 2 and 3).

“Pediatric surgery is the diagnostic, operative, and postoperative surgical care for children with congenital and acquired anomalies and diseases.”

2. The main objectives of this review are still unclear as to what the search strategy for the literature was, what keywords were used and what databases were examined. This is important information that any review should include.

As suggested by the reviewer, we have added a Methods section to explain the search strategy. (Page 5)

“A broad search of PubMed database was performed to identify articles that described pediatric surgery history, training, and simulation-based education. Search terms included “pediatric or paediatric” and “surgery” and “training” and “education” and “simulation”. Titles and abstracts were screened for content relative to the narrative review.“

3. The paragraph “1.1 History of pediatric surgery and training” is beyond the scope of this review. The focus of this review should be on the advances in simulation-based training in pediatric surgery. About 30% of the text of the article is a very detailed history which should be given in a short paragraph.

As suggested by the reviewer, we have shortened the history of training.

4. As we live in the era of minimally invasive surgery, much more attention and space should be given to minimally invasive training and the possible modalities. Minimally invasive simulators should be presented in much more detail.

As suggested by the reviewer, we have added more information on MIS to section 3.2 Widespread implementation of simulation-based curricula. (Pages 12)

“With the implementation of minimally invasive surgery three decades ago, MIS procedures have become more integrated into pediatric surgery training curriculum."

5. The authors should be examine the current situation in training the skills required for the different areas of pediatric surgery and compare them between Europe, Asia, and America…For example, the authors should comment on the current situation of training in minimally invasive surgery and compare the experience of trainees in paediatric surgery in Europe, America, Asia, etc.

As suggested by the reviewer, we have added the following to section 3.2 Widespread implementation of simulation-based curricula. (Pages 12 and 13)

“From 2004 to 2016, there has been a 30% increase in average number MIS cases per fellow in Canada and United States with variation in exposure among trainees. In contrast, 67% pediatric surgery trainees from European countries responded in a survey the challenges in their program performing MIS procedures and less than 5 out of 25 pediatric MIS procedures were performed by at least 50% of trainees. Like in Northern American, there was great variability in training and exposure in Europe as years spent in a general surgery department lead to greater number of higher complex procedures performed compared to years in pediatric surgery training.”

“Recently, Bailez et al has written about the development of minimally invasive surgery training program, onsite and telesimulated, with low-cost models in Argentina that has led to surgical efficiency and increase in complexity of cases performed.”

“As of today, some pediatric surgery training centers in Africa still have limited exposure to trauma, burn, and minimally invasive surgery.”

6. In addition, the development of instruments specifically for the paediatric population should be described, which the authors did not address.

Thank you for your comment. The following sentence was in section 2 Simulation in pediatric surgery training. (Page 5)

“For example, the peritoneal cavity in an infant is far smaller than an adult, and surgical instruments, hand movements, and strategies were made to accommodate for the restricted space.”

7. Finally, the review is very short (usually at least 4000 words are required, excluding references and explanations). Moreover, since many similar systemic reviews have been published on this topic, I do not see the scientific value of this simple narrative review.

Thank you for your comment. The purpose of this narrative review for the special issue of Advances in Pediatric and Neonatal simulation is to make both surgical and non-surgical providers more aware of surgical simulation based training in pediatric surgery.

Reviewer 5 Report

Comments and Suggestions for Authors

Dear Authors,

       Child and adult are two terms used to describe two stages of human beings in society and therefore, the main difference between the two words is related to the delimitation of the two stages. A child is a young person, possibly under the age of 18. An adult, on the other hand, is a fully grown human being. Adults in society have much greater responsibilities towards themselves and others. This is mainly due to their independent status. Children do not have a similar status because they are dependent on others and are still going through the socialization process. Therefore, it is impossible to say child = adult. Although it seemed like a sub-branch of general surgery until recently, pediatric surgery is a surgical department with its own methods.

     The unit that works in hospitals or other health institutions to treat urological problems and diseases that require surgical treatment of children between the ages of 0-16 is called pediatric surgery. In the pediatric surgery unit, diseases related to the respiratory and digestive systems of the body, chest, head, neck, trauma, endocrine surgery and gynecological disorders are treated. This unit deals with pediatric diseases, unlike general surgery. Additionally, the reason for establishing such a unit is that the methods used during the necessary surgical intervention on children may differ.

     The fact that both the anatomical and physiological characteristics of children are different from adults highlights the difficulties of this branch. In addition, while general surgery is divided into many sub-branches, the fact that pediatric surgery has not become specific is a part of these difficulties. The addition of congenital anomalies makes the incident even more important.

    Considering all these situations, the rapid division of pediatric surgery into branches, the fact that it is different from adult surgery, and the fact that the training is very broad and difficult make it necessary to carry out the basic training stated by the authors.

     My suggestion to writers; They can add to their articles about the difficulties of pediatric surgery training, that the anatomical area involved is wide but the working area is narrow, therefore it should be quickly divided into specific branches such as pediatric urology and pediatric thoracic surgery, and that they receive repetitive training both live and through simulations.

            CONCLUSION: Accept after minor revision

             Kind regards.

Author Response

Thank you for giving us the opportunity to submit a revised draft of the manuscript “Advances in Pediatric Surgery Based Training” for publication in special issues Advances in Pediatric and Neonatal Simulation for Children. We appreciate the time and effort that you  dedicated to providing feedback on our manuscript and are grateful for the insightful comments on and valuable improvements to our paper. We have incorporated most of the suggestions. Those changes are highlighted within the manuscript. Please see below, in blue, for a point-by-point response to the reviewers’ comments and concerns. All page numbers refer to the revised manuscript file with tracked changes.

My suggestion to writers; They can add to their articles about the difficulties of pediatric surgery training, that the anatomical area involved is wide but the working area is narrow, therefore it should be quickly divided into specific branches such as pediatric urology and pediatric thoracic surgery, and that they receive repetitive training both live and through simulations.

Thank you for your suggestion. There is a section 1 titled current challenges in training pediatric surgeons (Pages 4 and 5) and we have added to section 3.2 Widespread implementation of simulation-based curricula on the difficulties on MIS training in multiple regions and the establishment of programs designed to fill the void. (Pages 11-13).

We have added the following to Section 2 Onset of simulation-based training in pediatric surgery and the conclusion.

“Surgery in the pediatric patient requires operating in very small, narrow spaces of the thoracic, abdominal, and pelvic (genitourinary) compartments compared to adults.” (Page 5).

"From repetitive training involving both direct observation in the operating room to state of the art simulation-based training, today’s pediatric surgical trainees have a much broader exposure to pediatric diseases and procedures than their predecessors." (Page 14)

Round 2

Reviewer 3 Report

Comments and Suggestions for Authors

The paper is improved. The authors respected the comments.

Comments on the Quality of English Language

-

Reviewer 4 Report

Comments and Suggestions for Authors

The authors have considerably improved the manuscript. Although this report has no significant scientific value, it can be accepted in the present form as it provides a narrative overview of the current situation in pediatric surgery based on simulations.